# Impact of the COVID-19 Pandemic on Inpatient Antibiotic Consumption in Switzerland

**DOI:** 10.3390/antibiotics11060792

**Published:** 2022-06-11

**Authors:** Olivier Friedli, Michael Gasser, Alexia Cusini, Rosamaria Fulchini, Danielle Vuichard-Gysin, Roswitha Halder Tobler, Nasstasja Wassilew, Catherine Plüss-Suard, Andreas Kronenberg

**Affiliations:** 1Swiss Centre for Antibiotic Resistance (ANRESIS), Institute for Infectious Diseases, University of Bern, 3012 Bern, Switzerland; michael.gasser@unibe.ch (M.G.); catherine.pluess@unibe.ch (C.P.-S.); andreas.kronenberg@unibe.ch (A.K.); 2Division of Infectious Diseases, Cantonal Hospital Graubünden, 7000 Chur, Switzerland; alexia.cusini@ksgr.ch; 3Infectious Diseases and Hospital Epidemiology, Thurgau Hospital Group, 8596 Münsterlingen, Switzerland; rosamaria.fulchini@stgag.ch (R.F.); danielle.vuichard-gysin@stgag.ch (D.V.-G.); 4Institute of Hospital Pharmacy, Bern University Hospital, University of Bern, 3010 Bern, Switzerland; roswitha.haldertobler@insel.ch; 5Department of Infectious Diseases, Bern University Hospital, University of Bern, 3010 Bern, Switzerland; nasstasja.wassilew@insel.ch

**Keywords:** antimicrobial stewardship, inpatient antimicrobial consumption, COVID-19, surveillance

## Abstract

The aim of this study was to analyze inpatient antibiotic consumption during the first 16 months of the COVID-19 pandemic in Switzerland. The entire period (January 2018–June 2021) was divided into the prepandemic period, the first and second waves, and the intermediate period. In the first year of the pandemic, total overall inpatient antibiotic consumption measured in defined daily doses (DDD) per 100 bed-days remained stable (+1.7%), with a slight increase in ICUs of +4.2%. The increase in consumption of broad-spectrum antibiotics was +12.3% overall and 17.3% in ICUs. The segmented regression model of monthly data revealed an increase in overall antibiotic consumption during the first wave but not during the second wave. In the correlation analysis performed in a subset of the data, a significant positive association was found between broad-spectrum antibiotic consumption and an increasing number of hospitalized COVID-19 patients (*p* = 0.018). Restricting this dataset to ICUs, we found significant positive correlations between the number of hospitalized COVID-19 patients and total antibiotic consumption (*p* = 0.007) and broad-spectrum antibiotic consumption (*p* < 0.001). In conclusion, inpatient antibiotic use during the different periods of the COVID-19 pandemic varied greatly and was predominantly notable for broad-spectrum antibiotics.

## 1. Introduction

Beginning at the end of 2019, the COVID-19 outbreak has become a global health emergency and is putting an enormous burden on health care systems [1,2]. Confirmed cases of COVID-19 first appeared in Switzerland in February 2020. The first wave occurred in the spring of 2020, followed by a decline in cases during the intermediate period, and a second wave in the winter half-year of 2020/2021 [3,4]. Both waves led to elective procedures being postponed to maintain availability of hospital resources, especially intensive care beds, for COVID-19 patients. During these pandemic waves, Swiss hospitals treated up to 40% fewer patients than usual [5]. Studies from other countries also showed that hospital admissions in the first weeks of the COVID-19 pandemic were significantly lower for all major groups of non-COVID-19 patients, both for medical and surgical conditions, as well as for those conditions that would have required intensive care [6,7].

Although COVID-19 is a viral disease, the use of antibacterial treatments was a common practice for hospitalized patients, especially at the beginning of the pandemic. This was primarily due to the clinical uncertainty and lack of effective treatment options for SARS-CoV-2. Despite the low incidence of secondary bacterial infections, almost three-quarters of all COVID-19 patients were treated with at least one antibiotic, primarily at the beginning of the pandemic [8,9]. Initial studies of antibiotic consumption in other European countries showed increased antibiotic consumption during the pandemic, particularly of broad-spectrum antibiotics [10,11]. It was feared that overprescription and inappropriate consumption of antibiotics during the pandemic could exacerbate the already prevalent problem of antibiotic resistance [12,13].

The impact of the substantial number of COVID-19 patients and the marked decrease in non-COVID-19 patients in hospitals on inpatient antibiotic consumption has not yet been investigated in Switzerland. With this retrospective study, we aimed to measure inpatient antibiotic use, during and before the COVID-19 pandemic (2018–2021) in Switzerland. In addition, we aimed to analyze the potential correlations between the number of hospitalized COVID-19 patients and antibiotic consumption at hospitals that reported monthly data.

To our knowledge, this study is the first to examine inpatient antibiotic consumption across the different time periods of the COVID-19 pandemic and to evaluate the correlation between the number of hospitalized patients and antibiotic consumption.

## 2. Methods

### 2.1. Study Design and Data Sources

In this retrospective observational study, we analyzed annual data on inpatient antibiotic consumption from two data sources: (i) the IQVIA database including the amount of antimicrobials purchased in Switzerland by the hospital sector (Sell-in data, channel hospital), including acute care centers, rehabilitation, geriatric and psychiatric clinics, as well as nursing homes and (ii) ANRESIS sentinel network data, including annual antibiotic consumption data from 56 acute care hospitals, with 40 of these hospitals also providing ICU data (which are not available in the IQVIA dataset). Fifteen hospitals (including two university facilities) are located in French-speaking Switzerland, thirty-six in the German-speaking part of Switzerland (including two university facilities), and five hospitals in the Italian-speaking part of Switzerland. The ANRESIS monitoring data include 66% of the total number of acute somatic care hospital bed-days in Switzerland and 58%, 78%, and 90% in German-, French-, and Italian-speaking Switzerland, respectively.

The main analysis of monthly data also used these two datasets, monthly IQVIA data (January 2018–June 2021) and ANRESIS data available since January 2019 only from four hospitals, providing monthly data on hospitalized COVID-19 patients and antibiotic consumption. This study period was split into the prepandemic phase (January 2018–February 2020 for IQVIA or January 2019–February 2020 for ANRESIS); the “first wave” (March 2020–April 2020); the “second wave” (October 2020–May 2021); and two “intermediate periods” (May 2020–September 2020 and June 2021). Pandemic waves were defined according to data from the Federal Office of Public Health. Specifically, if the 14-day incidence rate over a month exceeded 120 COVID-19 cases per 100,000 inhabitants at least once and/or the number of hospitalized patients with COVID-19 was greater than 1000 patients per month in Switzerland, a month was considered a “wave month” [4].

For all analyses, only antibacterial agents for systemic use (J01) were included. The following substances were categorized as broad-spectrum antibiotics: meropenem, ertapenem, imipenem/cilastatin, aztreonam, cefepime, ceftazidime, piperacillin/tazobactam, and ticarcillin.

Sales data from the IQVIA database and consumption data from ANRESIS, expressed in the number of units or packages, were converted to defined daily doses (DDD) using the Anatomical Therapeutic Chemical Classification System (ATC/DDD) developed by the WHO Collaborating Centre for Drug Statistics Methodology [14]. For the IQVIA data, antibiotic consumption was expressed in DDD per 1000 inhabitants per day (DID) by using population estimates from the Swiss Federal Statistical Office for the calculation, and, for the ANRESIS data, in DDD per 100 bed-days by using the aggregated number of bed-days from all included hospitals. The monthly number of patients diagnosed with COVID-19 was collected for the entire hospital and the ICUs from four hospitals during the period March 2020–June 2021.

### 2.2. Statistical Analysis

To compare the annual antibiotic consumption data (total and broad-spectrum antibiotics only), consumption was calculated by linguistic region and year, including percentage changes from the previous year.

For monthly data, we used interrupted time series (ITS) analysis, which has been described as a useful method to show the impact of public health interventions [15,16]. The response variable was consumption in DID and consumption in DDD/100 bed-days for the IQVIA and monthly ANRESIS data, respectively. As an explanatory variable, a dummy variable, COVID-19, was created, representing the different periods of the COVID-19 pandemic. The predictor “time” (year/month) was integrated as a linear variable. For the COVID-19 dummy variable, a distinction was made among the prepandemic period (before March 2020), the first COVID-19 wave, the second wave, and the intermediate periods. A sine/cosine term was included in the model to describe seasonality in antibiotic consumption.

Pearson’s correlation analysis was used to test a possible correlation between the number of hospitalized COVID-19 patients and antibiotic consumption. All analyses were performed using the open-source software R (version 4.0.5).

## 3. Results

### 3.1. Annual Data

Using the ANRESIS dataset (which includes the number of bed-days, Appendix A Appendix A), we observed an increase in inpatient antibiotic consumption from 50.9 to 51.8 (+1.7%) DDD/100 bed-days from 2019 to 2020 (Figure 1A). Interestingly, we observed a −0.7% decrease in the Italian-speaking region, while consumption increased in the German- (+0.9%) and French- (+3.9%) speaking regions (Figure 1A). The same trends (total +4.2%, French +10.1%, German +3.4, Italian −3.5%) were observed when restricting analyses to only ICUs (Figure 1B). The consumption of broad-spectrum antibiotics increased homogeneously in all regions; for the entire hospital (total +12.3%, French +13.6%, German +12.2%, Italian +11.4%), as well as in only the ICUs (total +17.3%, French +22.3%, German +16.0%, Italian +10.6%) (Figure 1C,D).

Inpatient overall antibiotic consumption in DDD per 1000 inhabitants per day, calculated from the IQVIA^TM^ sales data, was comparable between 2018 and 2019, while decreasing from 1.56 to 1.45 DID (−6.5%) between 2019 and 2020 (Figure 2A). The decrease was most prominent in the German-speaking region (−8.7%) and more moderate in the French- (−1.0%) and Italian-speaking (−0.5%) regions (Figure 2A). In contrast, and consistent with the analysis of the ANRESIS data, the consumption of broad-spectrum antibiotics in DID increased an average of 10.2% from 2019 to 2020, with an increase of 21.4% in the French- and Italian-speaking regions and a smaller increase of 4.5% in the German-speaking region. (Figure 2B).

### 3.2. Monthly Data

Within the ANRESIS database, monthly consumption and occupancy data were available for four hospitals. Aggregated annual data for these hospitals were comparable to the entire, annual ANRESIS dataset (Appendix A). The segmented regression model revealed a higher consumption of all systemic antibiotics in the entire hospital in DDD per 100 bed-days during the first wave and slightly lower consumption in the second wave and the intermediate period (Figure 3A and Appendix A). However, none of these differences were significant. In a post-hoc analysis—comparing use of individual antibiotic groups—we found that the relative increase during the first wave compared to the same periods before the pandemic was highest for macrolides and carbapenems (Appendix A). (However, individual antibiotic groups were not included in the model.) Focusing on broad-spectrum antibiotics, we observed a significantly higher consumption of broad-spectrum antibiotics during the first wave (*p* = 0.030), while consumption was lower during the intermediate periods and approximately equal to previous years during the second wave (Figure 3B and Appendix A). Again, the differences between the second wave and the intermediate periods were not statistically significant.

Using the ANRESIS data, we restricted this regression analysis to ICUs. Again, we found significantly higher antibiotic consumption in the first wave (*p* = 0.023) (Figure 3A and Appendix A). In the second wave, consumption in the ICUs deviated only slightly from the expected trend, while significantly fewer antibiotics were consumed during the intermediate periods (*p* = 0.042) (Figure 3A and Appendix A). When focusing on broad-spectrum antibiotic consumption in the ICUs, we observed a significantly higher consumption during the first wave (*p* = 0.03), nonsignificant higher consumption during the second wave, and lower consumption in the intermediate periods. (Figure 3B and Appendix A).

Figure 4 illustrates the monthly distribution in DID of inpatient antibiotic sales between January 2018 and June 2021 (IQVIA data). Segmented regression analysis results show a significant increase in the consumption of all systemic antibiotics measured in DID in the first wave of the pandemic (*p* = 0.015), while a nonsignificant trend of decreased antibiotic consumption was observed in the second wave (*p* = 0.100) and the intermediate periods (*p* = 0.098) (Figure 4A and Appendix A). In the post-hoc analysis, the most prominent increase in the first wave was observed in macrolides, attributed mainly to the French-speaking region (Appendix A). For broad-spectrum antibiotic consumption, the interrupted time-series model revealed that monthly consumption measured in DID was significantly higher during the first wave (*p* = 0.027). It is noteworthy that actual consumption per month (individual points in Figure 4) was higher at the beginning of the second wave but fell back as the second wave progressed to the expected level of the previous year, resulting in an overall nonsignificant increase during the second wave (*p* = 0.289, Figure 4B).

### 3.3. Correlation between Antibacterial Consumption and the Number of Hospitalized COVID-19 Cases

In the correlation analysis, we included four ANRESIS hospitals for which both monthly data on antibiotic consumption and the number of patients with COVID-19 were available (Figure 5). Examining the entire hospital, we found no correlation between total antibiotic consumption and the number of hospitalized COVID-19 patients. However, we observed a significant, positive correlation between the consumption of broad-spectrum antibiotics and increasing numbers of COVID-19 patients (*p* = 0.018). Restricting the dataset to ICUs, we found significant positive correlations between the number of COVID-19 patients and overall antibiotic consumption (*p* = 0.007), as well as broad-spectrum antibiotic consumption (*p* < 0.001).

## 4. Discussion

Earlier studies on antibiotic consumption during the COVID-19 pandemic raised concerns that inappropriate antibiotic consumption in the treatment of COVID-19 patients could potentially contribute to the development and spread of antibiotic resistance [13,17]. The present study reveals that in Switzerland, the emergence of COVID-19 has not increased the overall consumption of antibiotics in inpatients. Antibiotic consumption measured in DID, based on 2020 antibiotic sales data from IQVIA even decreased by −6.5% compared with 2019. However, measuring antibiotic consumption in DID does not consider hospital occupancy, which (especially during the first wave) decreased in Switzerland due to postponing non-urgent treatments to spare ICU capacities [18]. Indeed, considering occupancy using DDD/100 bed-days, we found global inpatient antibiotic consumption to be relatively stable (+1.7%) in 2020 compared to 2019. These findings are similar to those reported by Grau et al. in a study with 66 hospitals in Catalonia, where they found a slight increase in total antibiotic consumption [19]. Interestingly, the same study found that total antibiotic consumption in the ICU decreased. This contradictory finding could be due to differences in ICU occupancy; while hospitals in Catalonia experienced a large increase in bed-days in the ICUs in 2020, the number of ICU bed-days in Switzerland was comparable to that in 2019 [17]. In contrast, similar results for the ICU as in Switzerland were also observed in other studies [10,20,21]. Considering only the consumption of broad-spectrum antibiotics, we consistently found an increase for DID (+10.2%), as well as for DDD/100 bed-days (+12.3%), as was found in other studies [11,22,23,24]. This increase in broad-spectrum consumption in DDD/100 bed-days was slightly more pronounced in the ICUs (+17.3%). These findings are consistent with studies showing that broad-spectrum antibiotics increased in hospitals during the first year of the COVID-19 pandemic [25]. Given the difficulty in distinguishing between viral and bacterial disease in critically ill patients and the initial uncertainty in treating COVID-19 patients, this increase can be well explained. Although COVID-19 did not affect all Swiss regions equally, differences at the regional level were small, with one exception. There was a slight decrease in overall antibiotic consumption in DDD/100 bed-days in the part of Switzerland most affected at the beginning of the COVID-19 pandemic, the Italian-speaking part. We attribute this difference to the absence of a university hospital in this region or possibly to a lower threshold for hospital admission (i.e., more hospitalized patients with only mild to moderate illness). Notably, there was a pronounced increase in macrolide consumption in the French-speaking region during the first wave. Francophone hospitals may have partially adopted the controversial recommendation of French experts to use hydroxychloroquine and azithromycin in COVID-19 patients [26].

The analysis of the monthly data clearly indicated that the COVID-19 pandemic should not be considered a homogeneous event but that the individual periods of the pandemic should be considered independently. During the COVID-19 pandemic, the evolution of antibacterial consumption fluctuated across the different waves. For example, antibiotic consumption was highest during the first wave, probably due to the uncertainty of how to treat COVID-19 patients [9], followed by a decline with the introduction of more specific treatment guidelines. This was also found in a study from Israel, where a continuous decrease in antibiotic consumption from wave to wave was observed [27]. However, especially for the sales data, an amplification of this effect by buying antibiotics in stock at the beginning of the pandemic cannot be ruled out. However, we do not believe that this is the only contributor since the same trends were also observed for the ANRESIS consumption data and at the beginning of the second wave. Except for the first wave, both the sales and consumption of all antibiotics across hospitals were below the predicted trend of the pre-COVID-19 period. However, when restricting the analysis to either ICUs or only broad-spectrum antibiotics, there was a clear increase in antibiotic consumption during both the first wave and the beginning of the second wave compared with the pre-COVID-19 period. This finding is consistent with a study showing that despite a relatively low prevalence of bacterial co-infections (3.5%) or secondary infections (14.3%) in patients with COVID-19, 58% of all hospitalized COVID-19 patients in high-income countries received at least one antibiotic [28]. Little has been published on antibiotic consumption during the different periods of the pandemic. In our opinion, the consumption differences may reflect changes in the patient profile during the pandemic periods, with fewer but predominantly severely ill patients during the first and second waves compared with the prepandemic or intermediate periods. On the other hand, it may also reflect behavioral changes in prescribing patterns. At the beginning of the pandemic, due to the high occupancy rate of ICUs and general uncertainty about the treatment of COVID-19 patients, treating physicians may have tended to prescribe broad-spectrum antibiotics leading to increased prescribing of these antibiotics. As the pandemic progressed and the treatment of COVID-19 patients became more routine, this initially high consumption was adjusted—an aspect that is also supported by a study by Henig et al. [27].

We also sought to better understand the opposing influences of an increasing number of COVID-19 patients and a decreasing number of other patients due to avoiding nonurgent hospitalizations and postponing elective interventions. We analyzed data from four ANRESIS hospitals that had data for both monthly antibiotic consumption and hospitalized COVID-19 patients. A significant positive correlation between the number of hospitalized COVID-19 patients and broad-spectrum antibiotic consumption was observed in the entire hospital and the ICUs, while in the entire hospital, we only observed a correlation for higher consumption of broad-spectrum antibiotics with increasing numbers of hospitalized COVID-19 patients. The results of this correlation analysis support our hypothesis that a relatively large number of broad-spectrum antibiotics were used, especially in the first phase of the pandemic, despite the low number of hospitalized patients. At the beginning of the second wave, consumption was comparable. However, the number of COVID-19 hospital admissions was higher, which better justifies the higher consumption of broad-spectrum antibiotics. To the best of our knowledge, no similar correlation analysis has been carried out, and comparable data in the literature are not available.

This study has some limitations. First, the IQVIA dataset included antibiotic sales to all Swiss hospitals, including nonacute care hospitals that specialized in psychiatry, rehabilitation, geriatrics, and palliative care, which did not treat patients with acute COVID-19 infection. Therefore, there may be an underestimation of the impact of the COVID-19 pandemic within this dataset. Second, the sales-only data do not directly reflect the actual consumption of antibiotics, which may lead to misinterpretations about antibiotic consumption across the different periods of the pandemic. For this reason, and to better reflect actual consumption, data from ANRESIS were also studied. However, the ANRESIS data are based on the number of packages dispensed by the hospital pharmacies to the different departments and not the actual quantity dispensed to patients. Patient-specific data were not available. In addition, although the ANRESIS dataset does not include all hospitals, the representativeness is quite high, with a distribution of hospitals across Switzerland (Appendix A) and coverage of approximately 66% of all bed-days in Swiss acute hospitals. For the monthly ANRESIS data, a regional analysis is impossible because the dataset included only four hospitals. However, we did not identify any obvious differences between these four hospitals and the entire ANRESIS dataset (Appendix A). Third, the model used in this study assumed a linear trend in outcomes within each segment, and this assumption of linearity may only hold for short time periods. Moreover, the short time periods for individual segments (e.g., the first wave had only two measurement points) made a statistically sound regression model difficult. Nevertheless, we believe that our data are valid. We found congruent results using different datasets, allowing for a differentiated and better understanding of the interactions between COVID-19 waves and antibiotic consumption in Switzerland.

## 5. Conclusions

Antibiotic sales to inpatient facilities decreased during the COVID-19 pandemic, while consumption, measured in DDD/100 bed-days, increased slightly. The impact of the COVID-19 pandemic on antibiotic consumption was most pronounced in the ICUs. The use of broad-spectrum antibiotics increased in hospitalized COVID-19 patients, which could potentially influence the development of bacterial resistance.

## Figures and Tables

**Figure 1 antibiotics-11-00792-f001:**
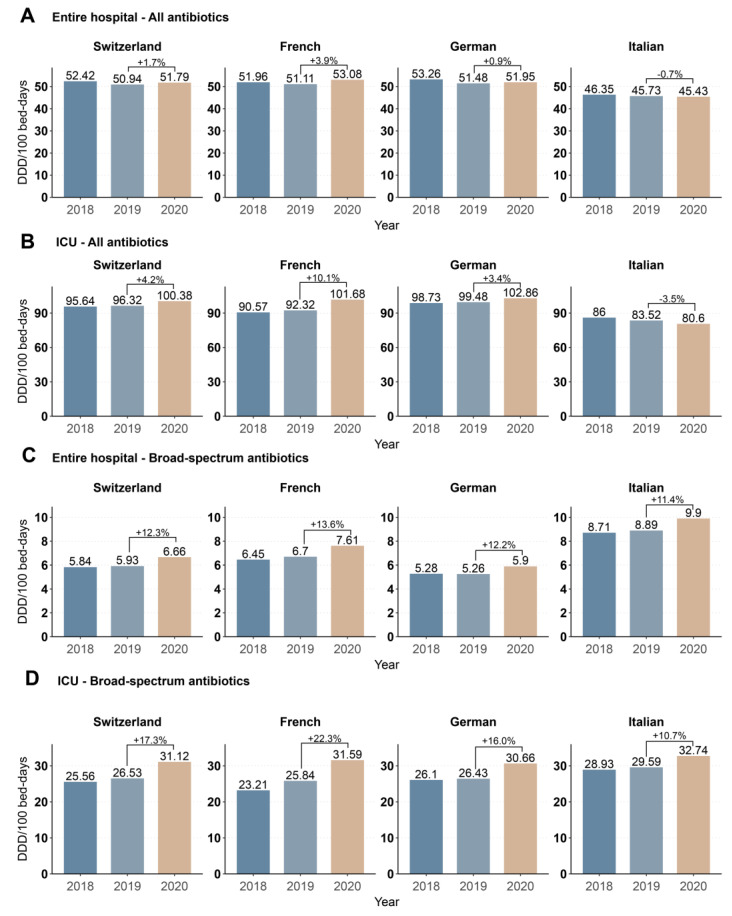
Antibiotic consumption in DDD/100 bed-days in Switzerland and the three linguistic regions from 2018–2020 for all antibiotics for systemic use (ATC code J01) in (**A**) the entire hospital and (**B**) ICUs and for broad spectrum antibiotics only, for (**C**) the entire hospital and (**D**) ICUs. Percentage changes between 2019 and 2020 are indicated.

**Figure 2 antibiotics-11-00792-f002:**
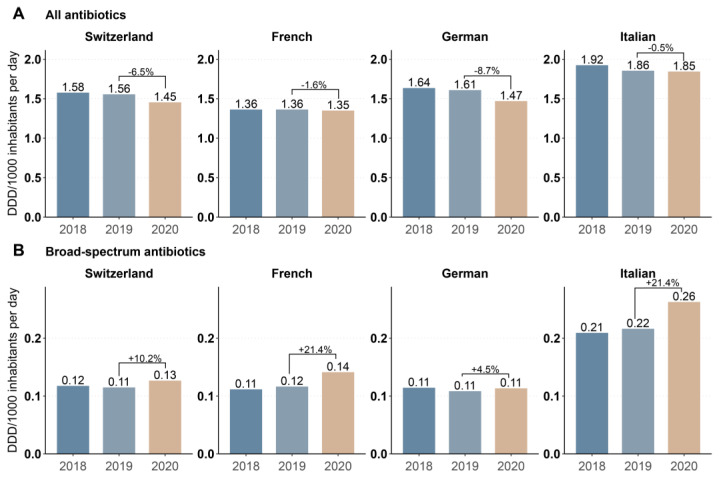
Annual antibiotic sales expressed in DDD per 1000 inhabitants per day for (**A**) antibiotics for systemic use (ATC code J01) and (**B**) broad-spectrum antibiotics in Switzerland and the three linguistic regions between 2018–2020. Data source: IQVIA sales data (Sell-in) from pharmaceutical industries to hospital sector.

**Figure 3 antibiotics-11-00792-f003:**
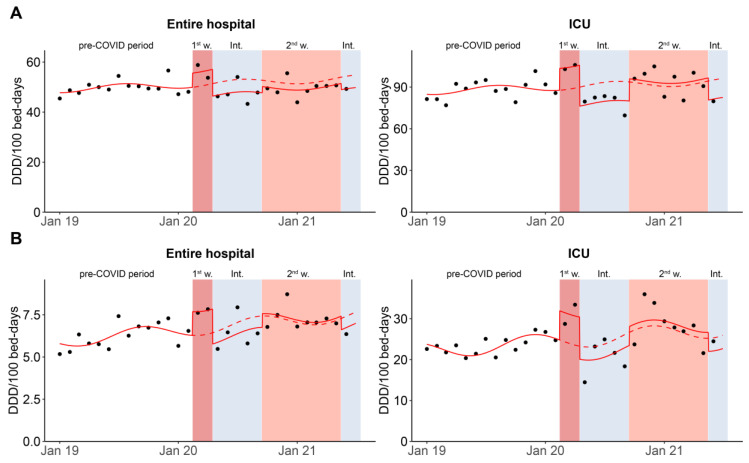
Analysis of monthly antibiotic consumption using the ANRESIS subset, measured in DDD per 100 bed-days for (**A**) all antibiotics for systemic use (ATC code J01) and (**B**) broad-spectrum antibiotics only, for the entire hospital and ICUs between January 2019 and June 2021. The graph shows the impact of the two waves of the pandemic (reddish background) on antibiotic consumption, based on a segmented regression analysis. The solid line shows the estimates of the segmented regression model. The dotted line is the estimated regression model assuming that the pandemic had not occurred. The dots show the effective consumption per month. ICU, intensive care unit; 1st w, 1st wave; Int., intermediate periods; 2nd w, 2nd wave.

**Figure 4 antibiotics-11-00792-f004:**
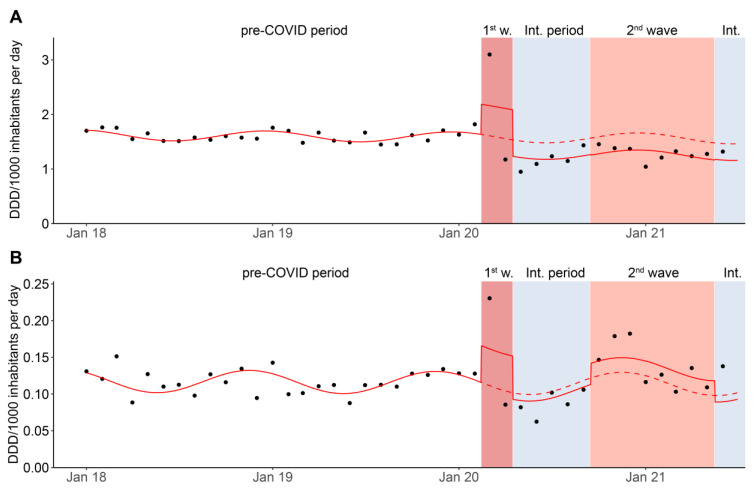
Monthly antibiotic consumption measured in DDD per 1000 inhabitants per day for (**A**) all antibiotics for systemic use (ATC code J01) and (**B**) broad-spectrum antibiotics between January 2018 and June 2021 (IQVIA data). The plot shows the impact of each pandemic phase (reddish background) on antibiotic consumption based on the segmented regression analysis. The solid line shows the estimates of the segmented regression model; the dotted line is the estimated regression model assuming the pandemic had not occurred. The dots show the effective consumption per month. Data source: IQVIA sales data (Sell-in) from pharmaceutical industries to the hospital sector. 1st w, 1st wave; Int. period., intermediate periods.

**Figure 5 antibiotics-11-00792-f005:**
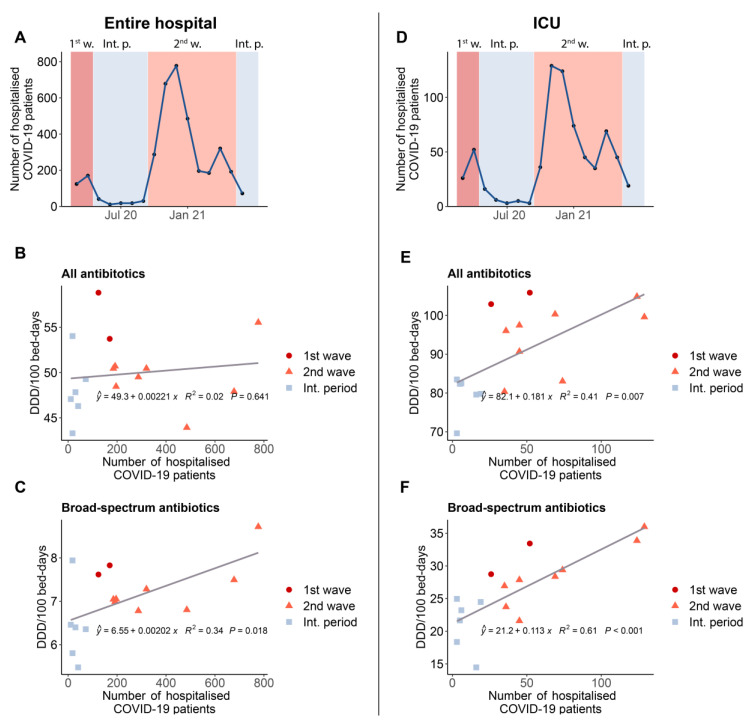
Correlation analysis for the entire hospital (**left** side) and ICUs (**right** side) based on aggregated data from four ANRESIS hospitals with available monthly antibiotic consumption and hospitalization data for COVID-19 patients. (**A**,**D**): Number of hospitalized patients with COVID-19 per month (**B**,**E**): Correlation between the consumption of all systemic antibiotics in DDD per 100 bed-days and the number of hospitalized patients with COVID-19 (**C**,**F**): Correlation between consumption of broad-spectrum antibiotics in DDD per 100 bed-days and the number of hospitalized patients with COVID-19. ICU, intensive care unit; 1st w, 1st wave; Int. p., intermediate periods; 2nd w, 2nd wave.

## Data Availability

The data are not publicly available.

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
