# Peer review of "Impact of the COVID-19 Pandemic on Inpatient Antibiotic Consumption in Switzerland"

_antibiotics, 2022, doi:10.3390/antibiotics11060792_

Round 1

Reviewer 1 Report

Authors used a longitudinal study to identify the impact of the COVID-19 pandemic on inpatient antibiotic consumption in Switzerland. Because antibiotic consumption is one of the key indicators in AMR countermeasures, it may be meaningful to analyze the impact of the COVID-19 pandemic. However, this article has not been fully answered some of questions due to the insufficient description.

First, authors show a lot of results in supplemental materials, but these results include major findings. For example, authors suggest “The increase during the first wave was proportionally highest for macrolides and carbapenems (Table S3).” and “The most prominent increase in the first wave was observed in macrolides, attributed mainly to the French-speaking region (Table S4).” using supplemental materials, but these results do not appear to be “additional” information, as authors mentioned in result section. Authors also show the result of Figure S1 in the result section using the same length as the result of Figure 1. Authors should add these figure and tables in the manuscript, but not supplemental materials.

Second, authors use data between 01/2018 and 06/2021 in figure 3, but the study period is between 01/2019 and 06/2021 in figure 2. This discrepancy may lead to readers’ distrust that authors may have deleted inconvenient data. Authors should use data from the same study period as much as possible, and if that is not possible, authors should justify why they do so.

Minor comment

P2. “Figure 1S A” and “Figure 1S B” may be typo of “Figure S1 A” and “Figure S1 B”.

Author Response

Reviewer #1:

Authors used a longitudinal study to identify the impact of the COVID-19 pandemic on inpatient antibiotic consumption in Switzerland. Because antibiotic consumption is one of the key indicators in AMR countermeasures, it may be meaningful to analyze the impact of the COVID-19 pandemic. However, this article has not been fully answered some of questions due to the insufficient description.

Major Points

First, authors show a lot of results in supplemental materials, but these results include major findings. For example, authors suggest “The increase during the first wave was proportionally highest for macrolides and carbapenems (Table S3).” and “The most prominent increase in the first wave was observed in macrolides, attributed mainly to the French-speaking region (Table S4).” using supplemental materials, but these results do not appear to be “additional” information, as authors mentioned in result section. Authors also show the result of Figure S1 in the result section using the same length as the result of Figure 1. Authors should add these figure and tables in the manuscript, but not supplemental materials

We fully agree with you about Figure 1S. Regarding the tables, we do not believe that it is beneficial for the understanding of the study to include the tables as well. We have deliberately chosen to focus the study on total consumption of all antibiotics and on broad spectrum antibiotics.  The analysis of individual antibiotics or groups of antibiotics was done post-hoc. In our view, it is therefore not appropriate to include the two quite detailed tables in the text when only a fraction of them are discussed in the results. The proposed ITS model for total antibiotics and broad-spectrum antibiotics from the main text cannot be applied to individual antibiotics and antibiotic categories without adaptations. Therefore, in Tables S3 and S4, the monthly mean values for the periods indicated are compared with the mean value of the same month for the periods before the COVID pandemic.

Second, authors use data between 01/2018 and 06/2021 in figure 3, but the study period is between 01/2019 and 06/2021 in figure 2. This discrepancy may lead to readers’ distrust that authors may have deleted inconvenient data. Authors should use data from the same study period as much as possible, and if that is not possible, authors should justify why they do so.

We agree that this is not ideally formulated. For technical reasons, it was not possible to extract the monthly data for 2018 for the hospitals. We have adjusted the corresponding section.

" The main analysis of monthly data also used these two datasets, monthly IQVIA data (01/2018-06/2021) and ANRESIS data available since 01/2019 only from four hospitals, providing monthly data on hospitalized COVID-19 patients and antibiotic consumption. "

Minor Points

  • “Figure 1S A” and “Figure 1S B” may be typo of “Figure S1 A” and “Figure S1 B”.

This has been adapted.

Reviewer 2 Report

Overall

-I would suggest to change the order of consecutive parts of the article beginning with introduction, then materials and methods, results and discussion.

Abstract

-A short sentence presenting the background for this study is needed to familiarize readers with the topic.

-I would suggest to reformulate the first sentence of the abstract – instead of writing what was done within this study, Authors could indicate what the aim of the study was.

-While using any abbreviation for the first time, Authors must explain it. What does “DDD” stand for?

-“During the first year of the pandemic, overall inpatient antibiotic consumption increased
slightly by 1.7% to 51.8 DDD/100 bed-days” – this is too vague. I suppose Authors are only talking about their study group so it should be clarified.

-“The increase in the consumption of broad-spectrum antibiotics was +12.3% for the entire hospital” – is it unknown what hospital we are talking about. Maybe you should add an information about where this study was conducted (what hospital), because as we know from Materials and Methods section, your study analyzed data from a number of Swiss hospitals.  

-Please add one sentence which will conclude the covered topic.

Introduction

-“Initial studies of antibiotic consumption showed increased antibiotic consumption during the pandemic, particularly of broad-spectrum antibiotics [8,9]. It was feared that overprescription and inappropriate consumption of antibiotics during the pandemic could exacerbate the already prevalent problem of antibiotic resistance [10,11].” – are you here talking about Switzerland? It should be clarified.

-“To our knowledge, this study is the first to examine inpatient antibiotic consumption across the different time periods of the COVID-19 pandemic in Switzerland” – what about other countries? Is any data from other countries available on this topic?

Results

-In the paragraph with annual data, Authors present information regarding the frequency of antibiotic consumption. However, it is unknown whether reported differences were statistically significant. Please conduct a proper statistical analysis to show p-values.

Discussion

-In this section you must compare your own results with other similar studies. There are many studies assessing the frequency of antibiotic use, for example:

https://www.ncbi.nlm.nih.gov/pmc/articles/PMC8472687/

https://www.ncbi.nlm.nih.gov/pmc/articles/PMC8703131/

https://www.ncbi.nlm.nih.gov/pmc/articles/PMC8603403/

References

-There are very few references. I think it should be valuable to add some new references.

Author Response

Reviewer #2:

Overall:

I would suggest to change the order of consecutive parts of the article beginning with introduction, then materials and methods, results and discussion.

We fully agree with you that the change of order is beneficial for the understanding of the publication, therefore we have changed that.

Abstract:

  • A short sentence presenting the background for this study is needed to familiarize readers with the topic.

We agree that it is appropriate to start the abstract with the aim of the study, which is why we have added the following sentence at the beginning:

"The aim of this retrospective observational study was to analyzed inpatient antibiotic consumption during the first 16 months of the COVID-19 pandemic in Switzerland."

  • I would suggest to reformulate the first sentence of the abstract – instead of writing what was done within this study, Authors could indicate what the aim of the study was.

Please refer to the previous answer.

  • While using any abbreviation for the first time, Authors must explain it. What does “DDD” stand for?

This item has been corrected.

  • “During the first year of the pandemic, overall inpatient antibiotic consumption increased

slightly by 1.7% to 51.8 DDD/100 bed-days” – this is too vague. I suppose Authors are only talking about their study group so it should be clarified.

               This sentence has been reformulated.

"In the first year of the pandemic, total overall inpatient antibiotic consumption measured in de-fined daily doses (DDD) per 100 bed-days remained stable (+1.7%), with a slight increase in ICUs of +4.2%."

  • Please add one sentence which will conclude the covered topic.

We agree with you, and have added a summary sentence to the abstract.

"In conclusion, inpatient antibiotic use during the different periods of the COVID-19 pandemic varied greatly and was predominantly notable for broad-spectrum antibiotics."   

Introduction:

  • “Initial studies of antibiotic consumption showed increased antibiotic consumption during the pandemic, particularly of broad-spectrum antibiotics [8,9]. It was feared that overprescription and inappropriate consumption of antibiotics during the pandemic could exacerbate the already prevalent problem of antibiotic resistance [10,11].” – are you here talking about Switzerland? It should be clarified.

This section has been adapted.

"Initial studies of antibiotic consumption in other European countries  showed increased antibiotic consumption during the pandemic, particularly of broad-spectrum antibiotics [8,9]. It was feared that over-prescription and inappropriate consumption of antibiotics during the pandemic could exacerbate the already prevalent problem of antibiotic resistance [10,11]."

-“To our knowledge, this study is the first to examine inpatient antibiotic consumption across the different time periods of the COVID-19 pandemic in Switzerland” – what about other countries? Is any data from other countries available on this topic?

In another extensive literature search, we found no publication that focused on antibiotic use in the different phases of the COVID-19 pandemic in a similar way as our study. There is one Dutch study that also addresses the different phases, but this is data on outpatient antibiotic consumption (https://pubmed.ncbi.nlm.nih.gov/35326772/). To be more precise, Switzerland was left out of this sentence. 

Results:

  • In the paragraph with annual data, Authors present information regarding the frequency of antibiotic consumption. However, it is unknown whether reported differences were statistically significant. Please conduct a proper statistical analysis to show p-values.

This is aggregated data from all participating Swiss hospitals. It does not have enough data points for a statistical analysis, therefore we remain descriptive.

Discussion:

  • In this section you must compare your own results with other similar studies. There are many studies assessing the frequency of antibiotic use, for example:

https://www.ncbi.nlm.nih.gov/pmc/articles/PMC8472687/

https://www.ncbi.nlm.nih.gov/pmc/articles/PMC8703131/

https://www.ncbi.nlm.nih.gov/pmc/articles/PMC8603403/

We agree that in certain parts we have not sufficiently reviewed our results in comparison with other publications. We have therefore discussed the data in more detail and have included two of the citations above.

References

  • There are very few references. I think it should be valuable to add some new references.

We have complemented the manuscript with references where it seemed reasonable to us.

Round 2

Reviewer 1 Report

Authors revised the manuscript, and I have no further comment.

Reviewer 2 Report

Authors have replied to all my suggestions and comments accordingly and what is important, they extended the discussion. Therefore, in my opinion this manuscript seems to be appropriate to be published in Antibiotics.